# Bio-Based Active Packaging: Carrageenan Film with Olive Leaf Extract for Lamb Meat Preservation

**DOI:** 10.3390/foods9121759

**Published:** 2020-11-27

**Authors:** Thamiris Renata Martiny, Vijaya Raghavan, Caroline Costa de Moraes, Gabriela Silveira da Rosa, Guilherme Luiz Dotto

**Affiliations:** 1Engineering Graduate Program, Federal University of Pampa, 1650, Maria Anunciação Gomes de Godoy Avenue, Bagé, Rio Grande do Sul 96413-172, Brazil; thamiris.martiny@hotmail.com; 2Chemical Engineering Department, Federal University of Santa Maria, Santa Maria, Rio Grande do Sul 97105-900, Brazil; guilherme_dotto@yahoo.com.br; 3Department of Bioresource Engineering, McGill University, 21111 Lakeshore Road, Ste-Anne-de-Bellevue, Montreal, QC H9X 3V9, Canada; vijaya.raghavan@mcgill.ca; 4Graduate Program in Materials Science and Engineering, Federal University of Pampa, 1650, Maria Anunciação Gomes de Godoy Avenue, Bagé, Rio Grande do Sul 96413-172, Brazil; caroline.moraes@unipampa.edu.br

**Keywords:** active packaging film, microwave-assisted extraction, *Olea europaea*, antimicrobial capacity, antioxidant activity, lamb meat

## Abstract

Carrageenan-based active packaging film was prepared by adding olive leaf extract (OLE) as a bioactive agent to the lamb meat packaging. The OLE was characterized in terms of its phenolic compounds (T.ph), antioxidant activity (AA), oleuropein, and minimum inhibitory concentration (MIC) against *Escherichia coli*. The film’s formulation consisted of carrageenan, glycerol as a plasticizer, water as a solvent, and OLE. The effects of the OLE on the thickness, water vapor permeability (WVP), tensile strength (TS), elongation at break (EB), elastic modulus (EM), color, solubility, and antimicrobial capacity of the carrageenan film were determined. The OLE had the following excellent characteristics: the T.ph value was 115.96 mg_GAE_∙g^−1^ (d.b), the AA was 89.52%, the oleuropein value was 11.59 mg∙g^−1^, and the MIC was 50 mg∙mL^−1^. The results showed that the addition of OLE increased the thickness, EB, and WVP, and decreased the TS and EM of the film. The solubility was not significantly affected by the OLE. The color difference with the addition of OLE was 64.72%, which had the benefit of being a barrier to oxidative processes related to light. The film with the OLE was shown to have an antimicrobial capacity during the storage of lamb meat, reducing the count of psychrophiles five-fold when compared to the samples packed by the control and commercial films; therefore, this novel film has the potential to increase the shelf life of lamb meat, and as such, is suitable for use as active packaging.

## 1. Introduction

Packaging systems occupy a prominent position in food processing and their use is indispensable for the distribution and commercialization of products on the market. Food packaging is increasingly influenced by the emergence of new technologies and new materials, such as active packaging [1]. This is a reflection of consumer demand for food that is original, underprocessed, and without chemical additives, and for more sustainable packaging materials [1,2]. Active packaging refers to packaging that modifies the condition of packaged food. Therefore, active packaging materials are used to extend a food’s shelf life and improve food safety [2,3]. Although extensive research is being carried out on active packaging, many have not yet been successfully tested on real systems. Thus, antimicrobial packaging in the form of biopolymeric films represents a viable alternative to active packaging [4,5].

From this perspective, new and unprecedented materials for bio-based packaging have been and continue to be developed, especially when it comes to biodegradable films [6], for example, biopolymers that are made from marine sources (chitosan, alginate) or raw materials from agriculture (corn and potato starches, zein) [7,8]. However, little is known about making films from a biopolymer obtained from red algae, namely, carrageenans, which is a group of complex biopolymeric substances called phycocolloids and represents a great prospect for the future [9,10]. The advantage of using carrageenan is that its gels have properties that alternate between liquids and solids, which allows for applications as additives in food manufacturing and the potential to form good biodegradable films due to their gelling power with excellent mechanical properties [11,12]. In addition, when making biodegradable films, a plasticizer is required. The plasticizer is a non-volatile substance that, when added to different systems, promotes physical and mechanical changes, allowing the films to become more malleable and resistant in terms of tension and elongation. Glycerol is a commonly used plasticizing agent in the production of films due to its compatibility and stability with biopolymer chains [13].

In order to transfer the additional antioxidant and antimicrobial properties to the films, the use of plant extracts is increasing [2] and olive leaf extract has characteristics that are suitable for this application. Olive leaves have substantial amounts of bioactive compounds (oleuropein, verbascoside, luteolin-7-*O*-glucoside, apigenin-7-*O*-glucoside, hydroxytyrosol, and tyrosol) in their composition, resulting in antioxidant and antimicrobial properties, which have been associated with their strong characteristics as preservatives [14,15,16]. The presence of phenolics and antioxidant activity in olive leaf extract has been reported [17,18]. Liu et al. [19] studied the antimicrobial activity of crude olive leaf extract and found that the extract nearly completely inhibited the growth of *Escherichia coli* and *Salmonella enteritidis*. They concluded that the extract could potentially be used to control pathogens in food products. Another advantage is that the leaves are a waste product of olive oil manufacture, where they make up an average of 10% by weight in the processing of olives [20,21].

The supplementation of olive leaf extract in biodegradable films has already been tested in some research [22,23]. Albertos et al. [22] produced fish gelatin films with the addition of olive leaf extract and used them as packaging for cold-smoked salmon; the films with the extract decreased the growth of *Listeria monocytogenes*. Bermúdez-Oria et al. [24] produced biodegradable films based on pectin and fish skin protein with the addition of olive extracts; the film preserved strawberries against mold during storage. It is also worth mentioning that two recent BR (Brazil) patents related to this research have shown the effectiveness of films produced with carrageenan and an incorporated olive leaf extract [25,26]. All information regarding the potential of reusing olive leaves shows their extracts as being promising as an ingredient with bioactive properties for applications in active packaging.

The effort to use less plastic and new packaging technologies is being mobilized around the world and is being felt even in the meat industry. Despite this, the development of meat packaging is a challenge because meat products often require materials with low transmission rates to extend and guarantee their useful life [7,27]. In this context, lamb meat is a suitable candidate for investment in protective packaging, as there are few studies on the changes in the quality of lamb meat during its storage. Lamb meat preservation methods include freezing, cooling, a vacuum, and a modified atmosphere [28], although the shelf life is limited by microorganism growth and lipid oxidation; however, these troubles can be solved by adding extracts of plants to the formulation of the packaging, making it active.

The potential applications for active antimicrobial packaging to extend the shelf life of meat and meat products were analyzed by Camo et al. [29]. Kuorwel et al. [30] studied synthetic and natural antimicrobial agents by incorporating them in films for packaging used basil, oregano, and thyme, as well as their essential oils. Given the above, the implementation of active packaging in the storage of lamb meat can be an innovative technology for the longer preservation of this meat and has the advantage that the extracts are not applied directly to the surface of the meat but are incorporated into the internal part of the packaging material [6,28,31].

Thus, the aims of this unprecedented research were to produce and evaluate the properties of an active carrageenan film containing olive leaf extract and to investigate its influences during the refrigerated storage of fresh lamb meat to produce an active packaging with the potential to replace films produced with synthetic polymers.

## 2. Materials and Methods 

### 2.1. Materials

#### 2.1.1. Olive Leaves

The *Olea europaea* L. type Arbequina was grown in southern Brazil (31°30′04.0″ S, 53°30′42.0″ W). After the collection, the leaves were oven-dried (ETHIK, Vargem Grande Paulista, Brazil) (40 °C, 24 h). Then, the leaves were ground (IKA^®^ A11BS32, Shanghai, China) and sieved (metal mesh 60, Metalúrgica Indústria Bertel, Caieiras, Brazil). Particles with a diameter of less than 0.272 mm were used.

#### 2.1.2. Chemicals

The carrageenan was purchased from Sigma-Aldrich (St. Louis, MO, USA), where the type was κ-carrageenan. The plasticizer used was glycerol, which was purchased from Mistura da Terra (Bagé, Brazil). The following reagents were bought from Sigma-Aldrich (St. Louis, MO, USA) and were of an analytical standard: Folin Ciocalteu’s phenol reagent, 2,2-diphenyl-1-picrylhydrazyl (DPPH), methanol, anhydrous sodium carbonate, gallic acid, and oleuropein. For microbiological analyses: nutrient broth, Müller–Hinton broth, PCA agar, and peptone (Himedia, Bengaluru, India), as well as sterilized distilled water, were used. 

#### 2.1.3. Bacterial Isolates

*Escherichia coli* ATCC 11229 was obtained from Oswaldo Cruz Foundation, Rio de Janeiro, Brazil. Before the tests, the bacterial culture was grown for 24 h in a nutrient broth at 35 °C.

### 2.2. Preparation of the Plant Extract

In order to obtain olive leaf extracts (OLEs), microwave-assisted extraction was performed in a multimode (closed) microwave unit (SCP Science, Baie-D’Urfe, QC, Canada) by applying the methodology of Rosa et al. [32]. The extraction was performed with 0.5 g of olive leaf powder in 25 mL of distilled water. Subsequently, the extracts were vacuum-filtered using Whatman^®^ Grade 4 filter paper (Sigma-Aldrich, St. Louis, MI, USA). Finally, they were lyophilized (Gamma 1–16 LSC, Christ, Osterode, Germany). Using the experimental optimization methodology (preliminary tests), the best extraction conditions were the following: 100 °C for 2 min at pH 6.

### 2.3. Extract Characterization

The total phenolics (T.ph) were determined using a method adapted from Singleton and Rossi [33]. The procedure was analogous to that of Martiny et al. [34]. The T.ph results were expressed in milligrams of gallic acid equivalent (GAE) per gram of dry matter. The analysis was performed in triplicate.

The method of Brand-Williams et al. [35] was used for the measurement of the antioxidant activity. A total of 0.2 mL of OLE was blended with 7.8 mL of DPPH (6 × 10^−5^ M) solution and stood still for thirty minutes at room temperature without light. The same procedure was performed with an aliquot of water in order to obtain control samples. The absorbance of the control and extract samples were measured using a spectrophotometer (Ultraspec1000, Amersham Pharmacia Biotech, Cambridge, England) at 517 nm. The free radicals captured by the DPPH were calculated using Equation (1) in triplicate:(1)$$A.\mathrm{activity}=\;\left(\frac{A_{C}-A_{\mathrm{OLE}}}{A_{C}}\right)\times 100\%,$$
where A.activity is the antioxidant activity expressed as a percentage (%), $A_{C}$ is the absorbance of the control, which was water, and $A_{\mathrm{OLE}}$ is the absorbance of the OLE samples. 

Quantitative analyses of the oleuropein in the OLE were performed using HPLC (Agilent 1100 series instrument, Santa Clara, CA, USA). Oleuropein was separated on a Discovery RP C18 column (Supelco, PA, USA; 5 μm, 25 cm × 4.6 mm) equipped with a Supelguard C18 cartridge (Discovery, St. Louis, MO, USA; 5 μm, 2 cm × 4 mm); it was then analyzed using a variable wavelength detector (VWD) that was set to 280 nm. The mobile phase was a mixture of water, acetonitrile, and acetic acid (80/19/1 *v*/*v*/*v*). The oleuropein was identified and quantified (mg∙g^−1^ of olive leaves (d.b.) (dry base) using an external standard and a calibration curve. 

The minimum inhibitory concentration (MIC) was evaluated against *E. coli* (ATCC 11229). *E. coli* was chosen because this bacterium is a potential causative agent of food-related diseases since meat is an environment that is favorable for its development. The MIC is the lowest concentration of the extract necessary for the complete inhibition of visible growth, where concentrations ranged from 5 to 150 mg∙mL^−1^ of the lyophilized extract. The antimicrobial activity was determined via inhibition analysis of the extracts obtained and was performed using the ELISA plate culture method adapted from [36,37]. In an ELISA microplate, 145 μL of sterile Mueller–Hinton broth, 135 μL of extract, and 20 μL of the culture containing the microorganism were pipetted into wells in triplicate. The antimicrobial activity was quantified using the absorbance (wavelength of 630 nm) difference between the two readings of the samples containing the extract in relation to the mean absorbance of the control samples in a spectrophotometer (PowerWave XS, Biotek, Winooski, VT, USA). Equation (2) was used to calculate the inhibition:(2)$$\mathrm{microbial}.A=\;\left[\;1-\left(\frac{A_{2\mathrm{OLE}}-A_{1\mathrm{OLE}}}{A_{2C}-A_{1C}}\right)\right]\times 100\%,$$
where microbial.A is the inhibition (%) and *A* is the absorbance, where the subindex “1” refers to the readings at 0 h, the subindex “2” refers to the readings at 16 h, the subindex “OLE” refers to samples containing extracts, and the subindex “C” refers to the mean absorbances for the control samples, which were the inoculants with water.

### 2.4. Carrageenan Biodegradable Films

#### 2.4.1. Film Preparation

The biodegradable films were formed according to the casting technique by Rosa et al. [23]. The proportion used for the film-forming solution was 1% (*w*/*v*) of carrageenan, 37.5% (*w*/*w*) glycerol (based on the carrageenan mass), and 62.5% (*w*/*w*) of OLE (based on carrageenan mass), which were dissolved in 50 mL of distilled water under agitation (1100 rpm) on a magnetic stirrer with heating (QUIMIS-Q261M23, Diadema, Brazil) at a temperature of 70 °C for 15 min. These conditions were established in preliminary tests. Initial studies were carried out to find the most suitable plasticizer and OLE concentrations. The results showed that films without a plasticizer were brittle, while those with high amounts of glycerol were stringy and difficult to remove from the plates. Thus, the concentration of 37.5% (*w*/*w*) glycerol was chosen for this research. It was determined that films made with OLE at concentrations below 62.5% (*w*/*w*) showed low or no antimicrobial effect; therefore, we chose to use a concentration of 62.5% (*w*/*w*) OLE for the formulation of the films. Films without the extract were produced as control films and called CAR-C, while films with the extract were called CAR-OLE. The filmogenic solutions were poured into 150 mm diameter acrylic plates and subjected to dehydration in an oven at 40 °C for 24 h. 

#### 2.4.2. Film Properties

With the support of a digital micrometer (Insize-IP65, São Paulo, Brazil), the measurements were taken at ten different positions of the film, thus detecting the mean thickness of the different samples of the biodegradable films produced.

The water vapor permeability (WVP) was ascertained using the ASTM standard E96/E96M methodology [38]. The mass gain, measured via anhydrous calcium chloride absorption, was monitored over ten days and the WVP was calculated using Equation (3):(3)$$\mathrm{WVP}=\;\frac{W}{t}\frac{e}{a\Delta P}\;,$$
where WVP is the water vapor permeability (g∙m^−1^∙Pa^−1^∙s^−1^), *W* is the absorbed moisture (g), *t* is the time (s), *e* is the thickness (m), *a* is the exposed film surface (m²), and Δ*P* the partial pressure difference (1176.17 Pa at 294.31 K).

The tensile strength (TS), elongation at break (EB) point, and elastic modulus (EM) of the films were measured using the ASTM standard D882-09 methodology [39]. The apparatus used was a Texturometer Analyzer (Stable Micro System TA.XTplus, Richmond, UK).

The film color was determined using the methodology proposed by Martiny et al. [34]. The color difference was calculated using Equation (4):(4)$$\Delta E^{*}=\sqrt{{\left(L_{C}^{*}-L_{OLE}^{*}\right)}^{2}+{\left(a_{C}^{*}-a_{OLE}^{*}\right)}^{2}+{\left(b_{C}^{*}-b_{OLE}^{*}\right)}^{2}},$$
where $\Delta E^{*}$ is the color difference (%); $L_{C}^{*}$, $a_{C}^{*}$, and $b_{C}^{*}$ are the color parameters of the CAR-C films; $L_{OLE}^{*}$, $a_{OLE}^{*}$, and $b_{OLE}^{*}$ are the color parameters of the CAR-OLE films. The *L** parameter ranges from 0 (black) to 100 (white). The *a** parameter measures the degree of red (+a) or green (−a) color and the *b** parameter measures the degree of yellow (+b) or blue (−b) color.

The water solubility of carrageenan-based films was determined using the method proposed by Gontard and Guilbert [40]. The film samples were initially dried to find the initial mass dry of the film samples. The samples were cut into 2 cm diameter discs, then immersed in 50 mL of water and the system received gentle agitation (100 rpm) at 20 °C for 24 h using a shaker incubator (SOLAB-SL 223, Piracicaba, Brazil). To determine the final amount of dry matter, the sample was dried (105 °C for 24 h). The solubility in water was calculated from the triplicate results using Equation (5):(5)$$\mathrm{WS}=\;\frac{m_{i}-m_{f}}{m_{i}}\times 100\%,$$
where WS is the water solubility (%), $m_{i}$ is the initial dry mass (g), and $m_{f}$ is the final dry mass (g).

### 2.5. Storage Study: Inhibition of Psychrophiles in Lamb Meat

To evaluate whether the biodegradable films of carrageenan with the olive leaf extract (CAR-OLE) had an influence on the housing of lamb meat, analyses of the growth of psychrophiles were made. Chilled lamb samples were purchased from the local market in Bagé, a city in southern Brazil. The tested meat was fresh and raw. The initial psychrophiles count was determined. Then, the samples were packed separately for each different film in duplicate, namely, CAR-C, CAR-OLE, and commercial polyvinyl chloride (PVC) film (Royal Pack, Alto Aririu, Brazil) (Figure 1). The packages were sealed using a heat-sealing machine (Lenor CP1-TH, ShenZhen, China) at 150 °C for 10 s. The packages were stored at 7 °C for 2 days. After the storage period, the final psychrophiles count was performed. The methodology used was the standard plate count [41] according to the American Public Health Association (APHA) [42]. The plate counts were made using a colony counter microprocessor (Electronics India, Parwanoo, India). The results were presented as the colony-forming units per gram of lamb meat (CFU∙g^−1^).

### 2.6. Statistical Analysis

The means of the experimental data and their respective deviations were calculated, where three replicates were performed. Significant differences between the means were determined via Tukey’s test at *p* < 0.05 using Statistica software (Stat Soft Inc., version 10, Tulsa, OK, USA). 

## 3. Results and Discussion

### 3.1. Olive Leaf Extract

Olive leaf is an excellent resource of phenolic compounds with an elevated antioxidant activity [32]. The results found in the current research also demonstrated these attributes. The total phenolic compounds (T.ph) were 115.96 ± 0.56 mg_GAE_∙g^−1^ (d.b.) and the result for the antioxidant activity was 89.52 ± 0.013%. Similar results for the antioxidant activity of up to 94% were found for olive leaf extracts obtained using a microwave [43,44]. The oleuropein content was 11.59 ± 0.004 mg∙g^−1^ (d.b.). In a study by Japón-Luján et al. [45], 23 mg∙g^−1^ (d.b.) oleuropein was extracted via the microwave technique using ethanol and water (80:20 *v*/*v*). In contrast, Khemakhem et al. [46] used the maceration technique and produced aqueous extracts of olive leaves, obtaining 2.65 mg∙g^−1^ (d.b.) of oleuropein. Jemai et al. [47] extracted 0.0432 mg∙g^−1^ (d.b.) of oleuropein from olive leaves using methanol with water (4:1 *v*/*v*) via maceration. Ansari et al. [48] extracted oleuropein with water at 60 °C and obtained 15 mg∙g^−1^ (d.b.). The values obtained in this research were comparable to the results reported in the literature cited. It is noteworthy that important differences were observed, such as the solvent type, extraction technique, and plant matrix origin. Another relevant fact was that in this research, water was used as a solvent (safe and non-toxic), which had a positive result in the oleuropein content obtained; as such, its use becomes competitive as a substitution for methanol and ethanol, which are commonly used in extractions.

Extracts of olive leaves produced via microwave-assisted extraction demonstrated antibacterial activity against *E. coli*, with an MIC value of 50 mg∙mL^−1^. This result is in accordance with the data previously reported by Pereira et al. [49], where they also observed an antimicrobial effect that was related to the concentration of OLE against bacteria and fungi. They found that the antimicrobial effect was attributed to oleuropein and the T.ph found in the extract, corroborating the results found in this research. Şahin et al. [43] similarly produced olive leaf extracts using the microwave technique but evaluated variables such as irradiation potency, irradiation time, and leaf mass. Their study revealed that the extract obtained under optimal conditions demonstrated an antibacterial effect against *Staphylococcus aureus*, with an MIC value of 1.25 mg∙mL^−1^.

### 3.2. Film Evaluation

Figure 2 illustrates the visual appearance of the biodegradable carrageenan films. The carrageenan-based biodegradable films were homogeneous, uniform, and could be easily removed from the plate. Films with no extract addition were clear and transparent and films with the extract addition were greenish-brown.

Table 1 provides the thickness, WVP, mechanical properties, solubility, and optical properties data of the formulated films. These results show that the incorporation of the OLE into the films caused an increase in the thickness. The thickness of the films increased linearly with the increasing mass; the addition of the extract in the film formulation caused an increase in the mass, which consequently caused an increase in thickness. Rhim [50] produced agar/carrageenan composite films (50:50) with incorporated clay nanocomposites; he obtained thicknesses of 0.0582 mm and 0.0643 mm as a result. The differences in the thicknesses found in the literature are due to the different methodologies used in the experimental preparation procedures, composition, and formulation of the film solutions.

There were significant differences in the WVPs of the films CAR-OLE and CAR-C, which was an indication that the water vapor permeability gradient was not the same for both films tested. The difference in the % relative humidity at the interfaces of both films was the driving force behind the diffusion of the water [51]. The WVP of a film depends on several variables, especially the thickness [52]. As there were differences in the permeability values between films with an OLE concentration, the increase in WVP may be due to the difference in the thickness. The increase in thickness in CAR-OLE was already expected due to the increase in the mass of the film-forming solution due to the incorporation of the extract. Films with OLE had a significantly higher thickness than those without OLE (Table 1). This observed effect may have been due to the interaction of the polysaccharides and OLE, which may have changed the saturation point. These interaction effects of the polymeric matrix and the different additives were also observed by Turco et al. [53], where they produced poly (lactic acid) thermoplastic starch films with cardoon seed epoxidized oil and observed that the higher the tension at the interface (less compatible mixtures), the greater the spaces between the phases, which favored the diffusion of water. Therefore, the OLE probably decreased the interfacial adhesion between the polymeric matrix and the dispersed phase, decreasing both the surface tension of the carrageenan and the intermediate space between the macromolecular chains, thus causing the acceleration of the diffusion of water molecules. Ideally, a film for the storage of lamb meat requires a low WVP; this result was found in this research and was corroborated by other studies that produced carrageenan films or films with the addition of OLE [22,23].

Comparing the data found in Table 1 with the properties of the commercial PVC film, which had a thickness of 0.004 mm, a WVP of 2.47 g∙m^−1^∙Pa^−1^∙s^−1^, elongation at break of 108.79%, and a tensile strength of 21.55 MPa based on data reported by Martiny et al. [34], it was concluded that the produced carrageenan films presented a superior thickness and WVP and inferior mechanical properties. These results suggest that the films can be improved in terms of their physical properties, where the investigation of biopolymeric blends for this purpose may be an option.

The addition of the extract in the composition of the films significantly changed the mechanical properties. There was an increase in the EB and a decrease in the TS. The fact that the biodegradable films of carrageenan prepared with the extract had a higher EB percentage may have been due to the fact that the extract had essential oils in its constitution since some studies show that lipid components can act as plasticizers in films. The addition of essential oils to the composition of biofilms caused a decrease in the rupture tension and an increase in the elongation of the films [54]. The EB is due to the alteration of the morphological structure of the film, which changes to a plastic flow regime up to the breaking limit. In this whole process, the chain alignment in the amorphous regions and the destruction of micro- and macro-crystalline structures are involved, with the consequent formation of fibrillar structures [55]. A polymeric material can undergo transformations in its original structure, as indicated by the EB test, and can limit the possible applications of the packaging [56].

Table 1 presents the values of the EMs for carrageenan films. The addition of OLE in the carrageenan films had an effect on the elastic modulus. There was a significant difference (*p* < 0.05) between both film formulations. The CAR-C film showed the highest EM value (40.50 MPa), which was in accordance with the result of the TS in this study. Films with higher EM data have turned out to be less flexible and more rigid than those with lower EM data [57,58]. The results found in this study showed that the OLE contributed significantly to the EM, producing more flexible films; this may have been due to the presence of lipid compounds in the extract that were able to form a continuous and cohesive structure. An EM result very similar to ours for a carrageenan film that was also plasticized with glycerol was found by Paula et al. [59], where the authors reported a value of 52.45 MPa. The results of this study were also in agreement with the study by Pereda et al. [60], which reported that the addition of olive oil into chitosan films enhanced the EM; according to the authors, this was because organic compounds, such as fatty acids and lipids, can contribute to the plasticization of the films. Therefore, the incorporation of the OLE had an effect on the elastic modulus of the carrageenan films produced in this study.

The addition of the extract to the film formulation did not cause a significant difference in solubility (Table 1). This high solubility is a feature of films formed from hydrocolloids, as they are highly hydrophilic, especially those made of polysaccharides and proteins [61]. In contrast, Rhim [50] only produced glycerol-plasticized carrageenan films, where the films completely disintegrated in water in just a 30 min test.

The results for the color parameters and color differences are shown in Table 1. Regarding the display of the carrageenan films, the addition of the extract decreased their moderate *L** value, which is an occurrence that was expected since the extract had a dark green color. The parameters *a** and *b** increased with the incorporation of the OLE, with both being positive, i.e., having mostly red and yellow components, respectively. The increase in *b** may have been linked to the phenolic compounds present in the OLE, which can absorb light of low wavelengths; similar behavior was obtained by Shojaee-Aliabadi et al. [62] for carrageenan films with incorporated essential oils. The films with oils differed from the control at 64.72%; these authors used glycerol plasticized carrageenan (1%) films of the same description as the CAR-C films, where *L** = 88.41, *a** = −0.27, and *b** = 0.86. Another relevant aspect is color, given that it interferes with the consumption profile. For the commercial PVC film, the optical parameters were *L** = 95.81, *b** = 1.088, and *a** = −0.14. As can be seen from Table 1, these data were close to the carrageenan films without the OLE, whereas for the films with the addition of the extract, the color parameters were significantly different. Despite the disadvantages attributed to the decrease in transparency, the addition of the extract has the advantage of decreasing oxidative processes caused by light on the food, though a more thorough investigation should be done regarding this feature.

### 3.3. Storage Study

The effects of the carrageenan film with or without the OLE on the development of psychrophiles microorganisms in the lamb meat packaging are shown in Table 2. The lamb meat was measured for initial psychrophiles and was determined to contain 1.10 × 10^5^ CFU∙g^−1^. The final concentration of psychrophile microorganisms in the lamb meat increased substantially during the 2 days of storage in all the samples. This trend was in line with the findings of Al Sheddy et al. and the EFSA BIOHAZ (European Food Safety Authority Biological Hazards) Panel [63,64], who demonstrated the ability of these microorganisms to develop in cool temperatures. It can be observed that during the storage period, the growth of psychrophile microorganisms was lower in lamb meat packed with the CAR-OLE film; this was approximately 5 times lower compared to the samples packed with CAR-C or the PVC film. CAR-OLE was able to slow the growth of psychrophiles compared to the control sample. The CAR-OLE efficiency can be attributed to its position on the surface of the lamb meat, where there was a higher microbial concentration. 

Some studies have also achieved successful results when designing active packaging for food applications; however, despite the advantages of this type of packaging, there are not many studies on lamb meat. In one study, Karabagias et al. [65] obtained a significant reduction in bacterial growth in lamb chops packed in a modified atmosphere packaging containing thyme oil. This active packaging extended its useful life by 2 to 3 days compared to that achieved with conventional modified atmosphere packaging. As opposed to using biodegradable polymers, Camo et al. [29] incorporated oregano extract into polystyrene (synthetic polymer) polymer matrix films and evaluated the application of this active packaging to the storage of lamb, successfully increasing the shelf life of the samples. Jancikova et al. [66] obtained edible carrageenan films with the incorporation of a lapacho extract; the results they obtained show that the films produced have the potential as a wrapped food commodity. However, they did not test the application in any food matrix. Saleh et al. [66] evaluated the effect of the direct use of olive leaf extract (not under a polymeric matrix like the present study) on the microbial growth of raw poultry meat; the results found by them revealed that the incorporation of olive leaf extract successfully reduced the microbial growth and the extract extended the shelf life of the poultry meat. 

Other research had already shown that OLE can inhibit the growth of microorganisms in in vitro assays. However, in this study, it was demonstrated for the first time that the carrageenan film with OLE could reduce the growth of psychrophile microorganisms in food, specifically in lamb meat. The application of the active carrageenan film with OLE provided an additional obstacle to the growth of unwanted microorganisms. The results reported in the literature prove that the union of a biopolymer, such as carrageenan, and olive leaf extract represents enormous potential regarding the development and real application as an active packaging for meat, which was corroborated by the promising results found in the present study.

## 4. Conclusions

From the aqueous OLE extraction process using the microwave-assisted extraction technique, an extract rich in total polyphenols was obtained, with high antioxidant activity and excellent oleuropein content, which was highly effective against *E. coli*. In addition, it has other benefits, such as low cost due to its source being a byproduct stream. When the OLE was incorporated into the biopolymeric matrix of carrageenan for the production of films, it decreased the growth rate of psychrophiles in packaged lamb meat and presented competitive physical characteristics relative to conventional polymeric matrices. The characterization of the biodegradable carrageenan films demonstrated that they were flexible and manageable, and that the addition of OLE significantly changed the thickness, the color parameters, the mechanical properties, and the barrier property. However, there was no significant change in the solubility. The carrageenan films with OLE were found to act as effective packaging that inhibited microbial growth in lamb. In order to avoid a significant effect on sensory properties, additional studies are recommended, including sensory tests on the product due to the application of the film.

## 5. Patents

There are two patents that resulted from the work reported in this manuscript.

Patent 1: Registry number: BR 102018013380-2 A2; publication date: 14/01/2020; title: Filme bioativo antimicrobiano à base de carragenana e extrato de folhas de oliveira.

Patent 2: Registry number: BR 102017013381-8 A2; publication date: 15/01/2019; title: Biofilmes antimicrobianos para proteção de alimentos.

## Figures and Tables

**Figure 1 foods-09-01759-f001:**
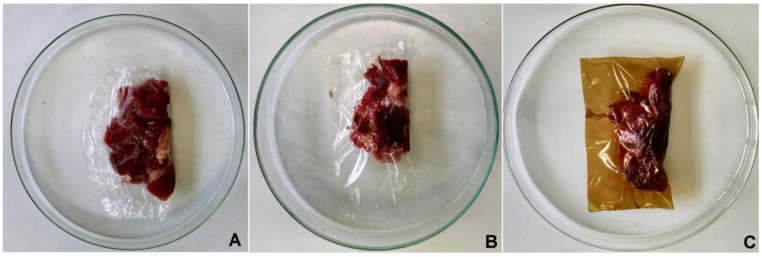
Samples of lamb meat that were packed using different films: (**A**) polyvinyl chloride (PVC) film, (**B**) carrageenan control film (CAR-C), and (**C**) carrageenan film with olive leaf extract (CAR-OLE).

**Figure 2 foods-09-01759-f002:**
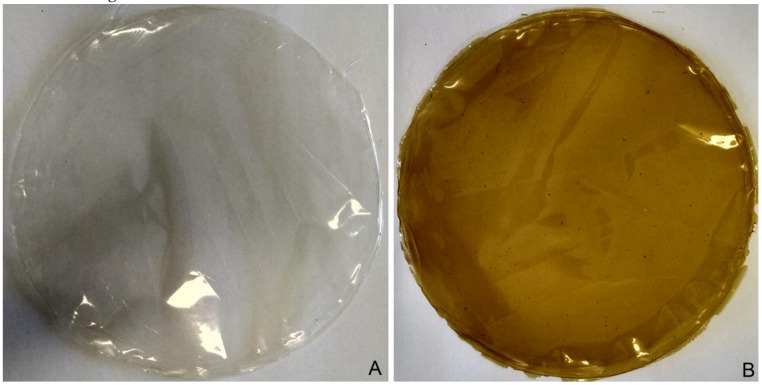
Visual appearance of the carrageenan films: (**A**) CAR-C and (**B**) CAR-OLE. CAR–C: carrageenan film, CAR–OLE: carrageenan film with olive leaf extract.

**Table 1 foods-09-01759-t001:** Thickness, water vapor permeability (WVP), mechanical properties, solubility, and optical properties of carrageenan films.

Physical Properties	CAR-C	CAR-OLE
Thickness (mm)	0.032 ± 0.004 ^a^	0.048 ± 0.004 ^b^
WVP (g∙m^−1^∙s^−1^∙Pa^−1^)	6.61 × 10^−11^ ± 1.6 × 10^−12 a^	7.43 × 10^−11^ ± 9.1 × 10^−13 b^
Elongation at break (%)	29.21 ± 0.12 ^a^	36.58 ± 1.70 ^b^
Tensile strength (MPa)	11.83 ± 0.23 ^a^	8.51 ± 0.09 ^b^
Elastic modulus (MPa)	40.50 ± 0.97 ^a^	23.34 ± 1.33 ^b^
Solubility (%)	82.60 ± 3.47 ^a^	76.60 ± 0.33 ^a^
**Optical Properties**	**CAR-C**	**CAR-OLE**
*L**	94.49 ± 0.21 ^a^	71.41 ± 0.92 ^b^
*a**	−0.145 ± 0.03 ^a^	7.78 ± 0.43 ^b^
*b**	3.32 ± 0.31 ^a^	63.26 ± 1.59 ^b^
ΔE	-	64.72

Data reported as the mean values ± mean deviation. CAR-C: carrageenan film, CAR-OLE: carrageenan film with OLE. $\Delta E^{*}$ is the color difference (%); *L**, *a** and *b** are the color parameters Different letters in the same line indicate significant differences between samples (*p* < 0.05).

**Table 2 foods-09-01759-t002:** The psychrophiles population measured initially and after 2 days of storage.

Samples Packed Using Different Films	Initial (CFU∙g^−1^)	2 Days of Storage (CFU∙g^−1^)
Fresh lamb meat	(1.10 ± 5.66) × 10^5^	-
CAR-C	-	(27.6 ± 8.66 ^a^) × 10^5^
CAR-OLE	-	(5.51 ± 7.62 ^b^) × 10^5^
PVC film	-	(23.9 ± 7.78 ^c^) × 10^5^

Data reported as the average values ± standard deviation. CAR-C: carrageenan film, CAR-OLE: carrageenan film with OLE, PVC film: polyvinyl chloride commercial film. Tukey’s multiple range tests were executed. Different letters in the same column indicate significant differences between the samples (*p* < 0.05).

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
