# Peer review of "Bio-Based Active Packaging: Carrageenan Film with Olive Leaf Extract for Lamb Meat Preservation"

_foods, 2020, doi:10.3390/foods9121759_

Round 1
Reviewer 1 Report
Authors developed here antibacterial and antioxidant biodegradable films based on carrageenan plasticized with glycerol and further incorporated with oil leaf extract (OLE) obtained by means of an aqueous extraction process. The work is well planed and organized. The obtained results are very interesting for meat packaging applications were a high flexibility is required. Authors not only characterize the materials obtained, but also studied them with real food systems.
My first concern is why only one formulation with OLE was developed. Why authors did not assayed different concentration of OLE in order to explore the influence of OLE on the mechanical, barrier and active performance of glycerol plasticized carrageenan-OLE films. The selection of these proportions should be clearly justified ore more samples should be added to the present research.
Abstract
- The polymeric matrix as well as the plasticizer used should be added in the abstract section.
- Escherichia coli should be in Italic
Introduction
Line 44-45: after authors mentioned that "Examples of these materials include biopolymers made from marine sources or raw materials originating from agriculture", some examples of these biopolymeric matrices have to be included to complete the state of the art, focusing in those reported developed biopolymeric formulations which are intended for meat products. As well, the advantages of using carrageenan as polymeric matrix instead of those biopolymeric formulations should be highlighted.
The interest of using glycerol as plasticizer should be added and also why it is used instead of other plasticizers should be explained.
Line 72-74: It is not clear if the intended use of the materials is to replace animal casings. If it is the main propose, it should be clear emphasized in this paragraph.
Materials and methods
Bacterial isolates: Escherichia coli should be in Italic.
Preparation of the plant extract: the number of Whatman filter paper and/or the pore size should be added.
In Equation 1 A. Active should be A. activity (%)
Line 137: In mg.mL-1, -1 should be a superscript.
In Equation 2 ?????????.? should be MIC (%), to better compare the results obtained here with those already reported in the literature.
Films preparation:
- according with the description reported here, if the final mass was 0.5 the plasticizer was used in 60 wt% with respect of the carrageenan matrix. It is very estrange to add more plasticizer than polymeric matrix, thus it should be interesting to better explain why this formulation was selected for the development of the materials. It is strongly recommended to express the proportion used with respect of the polymeric matrix in each material (i.e. carrageenan:glycerol:OLE XX:XX:XX or as wt% with respect of carrageenan matrix). In the same way, authors should explain why they have only developed one formulation with OLE and why this amount of OLE was selected.
- The dimensions of the acrylic plates should be added.
Line 170: (g.m–1.Pa–1.s–1) please put the superscript.
Line 201: in which local market were purchased the Chilled lamb samples. Please add city and country.
- Results and discussion
In Line 273 authors suggest that the reduction of L* value means a decrease on films. However, although transparency and lightness are very related properties, the decrease of L* means a decrease of lightness. Thus, authors should better explain the reduction on the transparency and if it is possible they should ad UV-vis measurements of the films.
Line 324:.. food commodity, however... should be: ...food commodity. However
Table 2. The ANOVA analysis should be added.
Line 291: Considering that only two formulations were developed "all films" should be changed by "both films".
Additionally, some WVP values of typically used meat biopolymeric films as well as carrageenan films should be added and compared with the results obtained here.
- Why authors did not assayed the antioxidant activity of films?
Conclusions
-The conclusions should be extended starting from the aqueous extraction OLE process.
-Line 335: E. coli should be in italic.
-Line 341: ... the barrier property; however... should be " the barrier property. However,..."
References:
the two mentioned patents in Patents section should be included as references and properly cited in the text (i.e.: introduction as well as materials and methods section)
Author Response
Please find in the attached document point-by-point response.

Reviewer 2 Report
Manuscript ID: foods-907854
Under entitled Title: Bio-based active packaging for preservation of lamb meat”. The author prepared an active packaging film using carrageenan - olive leaf extract. Then, evaluated the properties of olive leaf extract and carrageenan composite film. The efficiency of carrageenan - olive leaf extract films on lamb meat was tested.
- Here's, some suggested modifications.
- Title: I think it is better if the author makes the title of the manuscript more specific and show the main work of the manuscript like (Preparation of carrageenan/olive leaf extract- based active packaging film for lamb meat preservation).
- The abstract was not organized well. The author can have a look at this sequence ( Carrageenan-based active packaging film was prepared by adding olive leaf extract (OLE) as a natural antimicrobial and antioxidant agent. The effect of (OLE) on the thickness, water vapor permeability (WVP), tensile strength (TS), elongation at break (EB), color, solubility, and antimicrobial capacity of carrageenan film was determined. The results showed that……….(write the most significant results).
- Keywords: write meat
- Introduction part: Generally, introduction part was not well written and did not cover the main manuscript kaywords ( active packaging materials, carrageenan, olive leaf extract, meat preservation methods).
- Line 34-37: Rephrase the paragraph to be more clear.
- Line 38: alternatives packaging materials
- Line 44: Change ( Examples of these materials include) to For example, biopolymers ….
- Line 44-45: Give examples to marine sources or raw materials originating from agriculture. The manuscript missed some important points in material and methods as determination the total phenolic content and antioxidant activity for prepared films with or without olive leaf extract.
- Line 114: 6.10-5 M change it to 6×10-5 M.
- Line 141: What is the used device?
- Line 132: antimicrobial effect should be tested against at least one Gram-positive and Gram-negative bacteria.
- Line 154: correct this (filmogenic).
- Line 156: what is the quantity of used distilled water?
- I think that using high concentration from glycerol (about 60% based on carrageenan dry weight) and olive leaf extract (100% based on carrageenan dry weight) will give a film with very high elongation at break and too low tensile strength properties, this means that the film will be difficult to handle for storage of meat.
- Write the meaning of CAR-C, CAR-OLE, PVC film under table 2.
- What is the measurement unit of elongation at break?
- Results and Discussion part: Unfortunately this part unclear in many paragraphs many pieces of information missed.
Author Response

(The authors gave the same response as above.)

Reviewer 3 Report
The paper results interesting and useful for scientific community researching valid alternative to the exploitation of conventional plastics, mostly if the bioplastics can confer anmicrobial and antioxidant properties able to preverse food from spoilage and prolong their shelf-life by reducing oxidation. Aimed to this direction, the exploitation of olive extract from waste biomass as antimicrobial and antioxidant agent of carragenane bioactive packaging for lamb meat preservation, is absolutely valid and potentially applicable.
Rows:
28: potential
34: ...to preserve its products
35: remove "us"
45: remove ":"
46: phytocolloids
51: In addition, another major benefit of bioactive biodegradable films is that they can be used as a means of transmitting flavors, antioxidants, antimicrobials and dyes, which could be released to the food system during storage. Please, consider the above period
63: plastics not plastic
205: did you seal the packages? If yes, please describe the method.
248: Carrageenan-based biodegradable films were homogeneous, uniform and could be easily removed from the support (plate).
257-260: please simplifies this period
265: in Table 1, please add (%) in elongation at break
289: what about the elastic modulus? Together with tensile strength and strain at break, it represent the rigidity parameters. Please add this value to Table 1 to be more complete in tensile properties description
293: it is not properly correct what you say about thickness, since it is a normalizing factor of the water vapour trasmission rate. It is right what you mention before related to the difference in pathway of water depending on diffusion and permeation. To this aim, you could mention the following paper finding similar results well explained by considering both factors: Poly (Lactic Acid)/Thermoplastic Starch Films: Effect of Cardoon Seed Epoxidized Oil on Their Chemicophysical, Mechanical, and Barrier Properties
Rosa Turco, Rodrigo Ortega-Toro, Riccardo Tesser, Salvatore Mallardo, Sofia Collazo-Bigliardi, Amparo Chiralt Boix, Mario Malinconico, Massimo Rippa, Martino Di Serio and Gabriella Santagata, 2019. Coatings, 9, 574; doi:10.3390/coatings9090574.
Author Response

(The authors gave the same response as above.)

Round 2
Reviewer 1 Report
Authors have significant improved the manuscript. In my opinion it is now suitable for its publication in Foods
Author Response
Dear Reviewer,
We would like to thank you immensely for your evaluation and your time.
Kind regards,
The authors.
Reviewer 2 Report
Manuscript ID: foods-907854
Under entitled Title: Bio-based active packaging for preservation of lamb meat”. The authors revised the manuscript considering the reviewer's comments and there are some comments as following:
Line 21-22: Rephrase the following sentence (The formulation of films consisted of carrageenan, glycerol as a plasticizer and water as a solvent, in addition to OLE).
Line 28-29: Write the benefits of a film with a 64.72% difference color.
Line 29-30: Show the most powerful antimicrobial results.
Line 56: Could you mention the main role of plasticizers (e.g. it used for making the material softer)?
Line 107: what is the type of used carrageenan? γ-carr or κ-carr?
Line 174: Dehydration in a oven.
Line 223: Write the conditions (Time, Temp.) of sealing process, especially with biodegradable carrageenan films, and type of sealing machine.
Line 266: Remove support.
Line 270: Fig. 2, Change of Carrageenan films visual appearance to Visual appearance of carrageenan films.
Line 276-277: Rephrase the sentence that start with For carrageenan films…. again to be more clear.
Line 281: Table 1. please put the superscript letter at the end of number, e.g. from 0.032a±0.004 to 0.032 ± 0.004 a in all tables.
Also, resize the table to show all results together not in two separate parts.
Line 305: Rearrange the results of mechanical properties to be before the results of color, and to be the same sequence of results in the table.
- Where the discussion of Elongation at break (%) result.
- Generally, the result in table 1 need to rearrange in the followimg order (Thickness, Tensile strength, Elastic modulus, Elongation at break, WVP, solubility, then color result.
- arrange the result and discussion of Table 1 following the same result ordered in the table.
Line 326: add word of premeability after water vapor.
Line 328-329: Increase the thickness of CAR-OLE compared to control film may have resulted from an increase in the solid content of additives and the compatibility between film biopolymer and additives (there are many references that support this).
- Same situation in WVP property, which affect by the compatibility between film biopolymer and additives.
Line 333: Capital letters.
Reference: I think its too much references (more than 65 reference) and its better to reduce to 45-50.
Author Response
Please find the attached answers.
